# Agarose Hydrogels for Bone Tissue Engineering, from Injectables to Bioprinting

**DOI:** 10.3390/gels11040255

**Published:** 2025-03-28

**Authors:** Yibin Huang, Siyuan Peng, Yifan Chen, Bin Chu

**Affiliations:** 1School of Materials Science and Engineering, Xiamen University of Technology, Xiamen 361024, China; 2Key Laboratory of Biomedical Materials and Implant Devices, Research Institute, Tsinghua University, Shenzhen 518057, China

**Keywords:** agarose, hydrogel, bone, biological 3D printing technology

## Abstract

A great interest in agarose, with many health-promoting and gel properties, has been registered, especially in the field of bone regeneration and repair. Agarose and its major bioactive compounds are involved in biological activities such as inflammation, cell adhesion and proliferation, and the promotion of tissue repair. Due to its unique physical properties like gelation and solubility, agarose is increasingly utilized in the medical industry. The aim of this review is to present an overview of the applications of agarose hydrogels in bone tissue engineering, introducing agarose and its modified products as innovative solutions for bone regeneration. Additionally, the injectability of agarose hydrogels and their applications in bioprinting are also summarized. Data indicate that agarose will play an increasing role in current and future global medical sectors.

## 1. Introduction

The main constituents of bone, a porous mineralized structure, include blood vessels, cells, and calcified substances, particularly crystals of hydroxyapatite [1]. About 90% of the organic composition of bone is made up of non-collagenous proteins and type I collagen fibers, which make up the bone matrix, an essential part of bone [2]. Crystals of hydroxyapatite are firmly attached to collagen fibers, and their orientation matches that of the collagen. Osteocalcin, one of the non-collagenous proteins in the bone matrix, is essential for calcium binding, hydroxyapatite stabilization, and bone formation regulation. Bones provide a number of vital roles in the human body, such as supporting structure, shielding internal organs, promoting mobility, aiding in hematopoiesis, and preserving calcium–phosphorus equilibrium [3]. The quality of life for millions of people worldwide is impacted by bone tissue abnormalities and the rising incidence of bone disease brought on by the aging population [4]. Current treatments for bone defects fall into two main categories: non-invasive and invasive remedies [5]. Non-invasive remedies include biophysical stimulation, such as pulsed electromagnetic fields (PEMFs), which promote bone tissue regeneration through exogenous stimulation [6]. Conversely, invasive therapies include procedures like bone grafting [7] and bone tissue substitution [8]. Materials used for bone tissue engineering can be categorized as natural polymers, bioceramics, metals, and composites [9]. Ideal materials for bone tissue engineering should have the following conditions: good biocompatibility, sufficient mechanical strength, promotion of blood vessel formation, and the ability to induce differentiation of stem cells to osteoblasts [10]. However, not all materials can fulfill these requirements simultaneously.

Hydrogels stand out among many bone tissue engineering materials due to their three main advantages. First, they exhibit excellent biocompatibility, providing a suitable environment for cell growth and proliferation; second, hydrogels have remarkable cell adhesion, which not only further enhances cell growth but also facilitates cell adhesion to the material, which in turn contributes to tissue integration; and lastly, hydrogels have excellent water absorption, and they can absorb and maintain a large amount of water, thus simulating the hydration state of in vivo tissues [11,12,13]. Because of these characteristics, hydrogels are frequently utilized in tissue engineering and are perfect for creating artificial organs and tissues [14]. Of note are injectable hydrogels, which exist as a liquid before injection but transform into a solid gel after injection [15]. This property allows injectable hydrogels to effectively repair damaged tissues or organs in a minimally invasive manner. The repair process is facilitated by precisely injecting the hydrogel into the site of injury, where it cures to form a stable structure that provides support for cell growth and tissue regeneration [16].

Furthermore, injectable hydrogels are frequently utilized as bioink modules in bioprinting because of their soft, porous, and viscous properties [17]. Biological 3D printing technology takes advantage of the plasticity of hydrogels and their ability to form stable structures upon injection, allowing them to be deposited and molded into complex three-dimensional tissue structures in a precise manner. During the bioprinting process, injectable hydrogels can contain cells, growth factors, or other bioactive molecules, providing a powerful tool for tissue engineering and regenerative medicine [18].

Agarose hydrogels are widely used in drug delivery and tissue engineering due to their thermoreversible gel properties, excellent mechanical properties, good biocompatibility, and low economical cost [17,19]. In addition, agarose has a high molecular weight and branched structure, allowing its side chain atoms to form hydrogen bonds with water and themselves. During gelation, agarose creates a helical structure and cross-links via hydrogen bonds, forming a 3D network. This enables the formation of agar hydrogels without toxic cross-linking agents, enhancing their appeal as biocompatible polymers [20,21]. Nowadays, the use of agarose hydrogels as bioinks in bioprinting technology offers a range of innovative solutions for bone tissue repair and regeneration.

## 2. Agarose Hydrogel

Agarose is a linear natural polymer widely found in red algae [22]. As shown in Figure 1, its main components are agarose polymers, which are formed by alternating linkages of repeating D-galactose and 3,6-endoethergalactose (3,6-Anhydro-L-Galactose), a structure that confers on the agarose-based hydrogel an adequate gelling ability even at low concentrations [23]. The physical characteristics of agarose can be controlled, and the gel’s hardness, elasticity, and porosity can be changed to mimic the mechanical characteristics of different tissues by varying the agarose concentration [24]. In addition, agarose is able to form solid gels upon cooling in water; this gelation process is reversible and can be melted again into liquid form upon heating [8] (Figure 2). Agarose plays an important role in a variety of cellular behaviors, including the ability to promote cell migration, proliferation, and differentiation, as well as a role in angiogenesis [25]. Mechanistically, agarose hydrogels reduce inflammatory responses by creating a hydrated microenvironment that limits excessive immune cell infiltration, while their neutral charge and polysaccharide backbone may inhibit the activation of pro-inflammatory pathways such as NF-κB [17]. This immunomodulatory capacity, combined with its ability to accelerate wound healing and influence morphogenetic processes, underscores agarose’s multifunctionality in bone regeneration. Moreover, it can participate in the regulation of inflammatory responses, accelerate wound healing, promote tissue repair, and influence morphogenetic processes [26]. Hydrogels have a three-dimensional cross-linked polymer network, a structure that allows them to absorb and retain large amounts of water while maintaining a certain shape and stability. Hydrogels are soft and compliant, adapting to the shape and mechanical properties of the surrounding tissue. In addition, the porous structure of hydrogels provides a large amount of space that facilitates cell migration, nutrient transport, and metabolic waste elimination [27]. Although agarose hydrogels provide a simple, reliable, and low-cost, three-dimensional culture environment capable of supporting the growth and differentiation of human bone marrow cells [28,29], it is important to note that polysaccharides like agarose may activate the complement system via surface-exposed hydroxyl groups, potentially triggering inflammatory responses in vivo [20]. However, the major drawback of agarose hydrogels is the lack of cell adhesion, which is mainly due to their lack of specific cell adhesion sites [30,31]. Furthermore, agarose-formed hydrogels usually have high rigidity and low surface roughness, whereas cell adhesion and proliferation usually require a certain surface roughness and a specific topology [32]. Scientists have enhanced the cell adhesion of agarose hydrogels by various methods such as chemical modifications to bind specific cell adhesion peptide sequences (such as GRGD sequences columns) to agarose [33], mixing with other materials to form composite hydrogels [34] and the addition of growth factors [35] to the hydrogels. Next, we will outline the preparation of agarose hydrogels in bone tissue engineering and common strategies to overcome their drawbacks.

### 2.1. Agarose-Ceramic Composites

In order to be used in clinical settings, bone engineering materials must meet a number of stringent criteria. These include good biocompatibility and degradability to ensure that the material is non-toxic to cells and tissues and can be absorbed by the body; sufficient mechanical strength and elasticity to withstand physiological stresses and adapt to dynamic environments; moderate pore size and connectivity to facilitate cellular infiltration, distribution, and re-vascularization; and high drug-loading capacity, controlled release, and stimulus responsiveness for effective drug therapy [36]. However, from a materials science perspective, it is often difficult for a single material to combine all the desired properties. Therefore, researchers have worked on the design of composite materials aiming to incorporate the advantages of different materials [37]. Ceramic materials as biomaterials in the biomedical field, especially in orthopedic and dental applications, exhibit multiple significant advantages that make them of high clinical value. Among them, hydroxyapatite (HA) has been widely used in bone tissue engineering due to its high chemical similarity to bone and teeth [38]. This similarity not only endows HA with excellent biocompatibility but also enables it to form a good bond with host tissues and promote bone regeneration and repair. As a result, HA is an ideal choice for bone repair and substitution materials, and it has been widely used in the fields of bone defect filling [4], bone fracture fixation [39], and dental implants [40], which have significantly improved the therapeutic efficacy and the quality of patient recovery.

#### 2.1.1. Hydroxyapatite/Agarose Composites for Bone Defect Repair

In 2003, Tabata et al. formed HA in agarose gels through an alternating immersion process, a process that not only promotes gelation but also enhances the cross-linking of agarose gel and promotes apatite formation. The HA/agarose gel was able to stop bleeding within seconds compared to pure agarose hydrogel; furthermore, the HA/agarose gel was completely resorbed and replaced by newly formed bone tissue within 12 weeks after implantation, and inflammatory cells were not observed around the material [41]. The investigators also compared the effects of different HA concentrations on non-hypertrophic and hypertrophic chondrocytes cultured in composite scaffolds. In 2012, Khanarian et al. designed and optimized a hydrogel–ceramic composite scaffold consisting of agarose and hydroxyapatite (HA) for the regeneration of osteochondral interface. The addition of HA had no significant effect on the biosynthesis and hypertrophy of deep-zone chondrocytes (DZCs), but hypertrophic chondrocytes showed a higher potential for matrix deposition and mineralization in the presence of HA. The ceramic phase of the scaffolds was optimized in terms of particle size (200 nm vs. 25 mm) and dosage (0–6 *w*/*v*%.). It was also demonstrated that the addition of HA significantly improved the compressive and shear mechanical properties of the scaffolds. Scaffolds with 3% micron-sized HA showed the best performance in terms of matrix content, mechanical properties, and mineralization potential. In particular, the best performance in terms of matrix content, mechanical properties, and mineralization potential was demonstrated [42]. In 2015, Iwai et al. used HA/agarose gel for the repair of alveolar bone defects due to cleft lip and palate. HA/agarose gel was prepared by alternating immersion technique and applied to the alveolar bone defects by mixing it with an equal volume of autogenous bone. The results showed that the bone density at the graft site increased and remained stable at 1 month postoperatively. Complications such as infection and long-term inflammation of the grafted bone did not occur with the use of HA/agarose gel. The mixture of HA/agarose gel with autogenous bone was as effective as autogenous bone in the treatment of bone defects due to cleft lip and palate and reduced the amount of autogenous bone required [43]. This proves once again that HA/agarose gel can be used in bone tissue engineering to effectively help bone tissue repair. In order to further improve the mechanical properties of the scaffold and promote the proliferation and activity of bone marrow mesenchymal stem cells (MSCs). In 2019, Luo et al. prepared a three-dimensional N-doped graphene-hydroxyapatite/agarose (AG/NG-HA) composite bioscaffold for enhanced bone regeneration by a simple hydrothermal/crosslinking/freeze-drying method. In in vitro experiments, AG/NG-HA scaffolds showed good cell adhesion, enhanced alkaline phosphatase activity, and mineralization capacity [44]. Antimicrobial peptides have the ability to prevent bone infection; Mario Mardirossian et al. combined antimicrobial peptides with biocompatible scaffolds. The advantages of agarose and alginate in the preparation of polysaccharide/hydroxyapatite porous bone scaffolds containing the proline-rich antimicrobial peptide B7-005 were compared. It was found that alginate was initially selected to utilize its negative charge for the efficient loading of B7-005, but strong interactions with the positively charged peptide hindered its release. Therefore, alginate was replaced by agarose to prepare scaffolds with similar structural, porosity, and mechanical properties. Furthermore, the agarose scaffolds could release B7-005 within 24 h without affecting MG-63 cell adhesion and proliferation, and the cell proliferation could last for two weeks. However, B7-005 was effective against some bacteria but ineffective against others, such as *S. aureus* and *P. aeruginosa*, and it needs to be further optimized to enhance the antimicrobial effect [45]. The combination of hydroxyapatite and agarose has significant potential for application in the field of bone tissue engineering; however, there is still room for further optimization in the gel preparation process.

In order to accelerate the preparation process, in 2007, Watanabe et al. used an innovative electrophoretic technique that enables rapid preparation of HA/agarose gels. This composite has an interconnected structure in which the HA particles have a diameter of about 1 µm, and the total number of HA particles was estimated by quantitative analysis of calcium ions. In 1 mg of dry composite, 10 µg of HA was formed. In addition, the HA particles have an amorphous structure, which means that these particles are more soluble under physiological conditions compared to more crystalline HA. Rapidly formed HA/agarose gel composites may be good candidates for injectable biomaterials, especially in the field of orthopedic, oral, and maxillofacial surgery [46]. Furthermore, in order to avoid random dispersion and agglomeration of inorganic particles in conventional methods and to ensure the homogeneity and consistency of the nanocomposites, Jingxiao Hu et al. successfully prepared homogeneous HA/agarose gel nanocomposites for weight-bearing bone tissue substitution by a novel in situ precipitation method. The spherical inorganic nanoparticles (~50 nm) in this nanocomposite were uniformly dispersed in an organic matrix, and the crystalline regions were tightly bound to the amorphous regions. The agarose forms a double helical conformation during cooling, forming a three-dimensional network structure through intermolecular hydrogen bonding, and this structure provides a stable framework for the nanocomposite and enhances the mechanical properties of the material. Such uniformly dispersed HA nanoparticles significantly improved the mechanical properties of agar/HA nanocomposites with the highest modulus of elasticity up to 1104.42 MPa and the highest compressive strength up to 400.039 MPa, which showed potential as a potential load-bearing bone replacement material [47].

HA/agarose composites are widely recognized for their superior mechanical strength and osteoconductivity, making them an ideal material for load-bearing bone defects. Their ability to rapidly induce hemostasis and integrate with host tissues underscores their clinical utility. However, their slow degradation kinetics may delay bone remodeling in dynamic physiological environments, and their lack of inherent antimicrobial activity limits their application in infection-prone sites. Compared to β-TCP/agarose scaffolds, HA composites provide long-term structural support but sacrifice rapid osseointegration. In contrast, β-TCP/agarose systems effectively address these limitations, as will be discussed in the following subsection.

#### 2.1.2. β-TCP/Agarose Scaffolds for Rapid Osseointegration

β-TCP ceramics are considered to be a resorbable biomaterial due to their high absorption rate. The absorption process of TCP involves chemical solubilization and cell-mediated uptake [48]. In contrast, β-TCP is more soluble, and HA is relatively stable in body fluids. It was noted that HA dissolves too slowly in the human body, while β-TCP dissolves too quickly, both of which are detrimental to osseointegration. Therefore, the focus of research has shifted to biphasic calcium phosphate ceramics (BCPs) composed of HA and TCP [49]. Potoczek et al. prepared highly porous HA-β-TCP bioceramic foams by a gel casting technique, using agarose as a solidifying agent. The macroscopic porous microstructure of HA-β-TCP foams typically consists of approximately spherical pores (cells), which are connected to each other through circular windows. The foams show a broad pore size distribution, with cell and window pore sizes ranging from 250 to 900 μm and 25 to 250 μm, respectively. The plurality of spherical pores was about 500 µm, while the plurality of windows was about 100 µm. In addition, a small amount of wall micropores was confirmed by SEM and Hg porosity analyses, with pore sizes ranging from 0.2 to 0.9 µm. Porous (P = 90%) HA-β-TCP scaffolds with two types of macropores and a small amount of micropores were obtained [50]. In 2008, Sánchez-Salcedo et al. designed architectures to prepare BCP-agarose macroporous scaffolds with accurately designed and controlled pore structures by an innovative low-temperature molding method in combination with stereolithography for bone tissue engineering. Robust dry scaffolds with high porosity (total porosity: 80%), a pore size distribution of 70 μm, thoroughly open pores, and customized pore sizes were obtained. This provides more attachment sites for cells, enhances the exchange of nutrients and metabolic wastes, and makes it easier for cells to adhere to the surface and internal pores of the hydrogel. These scaffolds showed flexibility in contact with body fluids and were able to exert pressure on the patient’s bone defects [51]. The compressive behavior of BCP–agarose composite scaffolds in bone regeneration was investigated by Puértolas et al. in 2011 [52]. Using both static and cyclic compression tests, the study assessed the mechanical characteristics of dense and designed porous architectural bioceramic scaffolds based on the BCP system and BCP-agarose system. The results showed that the dense and designed porous architecture scaffolds in the BCP system exhibited brittle behavior. Agarose provided toughness, ductility, and rubbery consistency to the ceramic BCP–agarose system, allowing it to maintain its initial cylindrical structure at strains up to 60%, with maximum strengths of 10–50 MPa. This combination of ceramic and organic matrix helps avoid the inherent brittleness of bioceramics and enhances the compression resistance of hydrogels. The study also observed, for the first time in bioceramics and bioceramic-agarose systems, mechanical behavior similar to the Mullins effect in carbon black-filled rubber systems [52]. To further promote new bone formation and increase fibroblast density and neoangiogenesis, in 2019, Hasan et al. prepared and evaluated a novel BCP-CSD–agarose composite microsphere for bone tissue engineering. In this study, for the first time, such composite microspheres were prepared by a one-step method, which contained BCP (20 wt%) and calcium sulfate dihydrate (CSD, 20 wt%), enhanced with agarose (1 wt%). It was found that this composite microsphere was able to rapidly release Ca^2+^ ions in simulated body fluid (SBF), thereby increasing interaction with the microenvironment and supporting cell adhesion and proliferation. Compared to CSD-free composite microspheres (SA-1), CSD-containing composite microspheres (SA-2) showed a faster degradation rate and acidic pH in SBF, which helped promote bone regeneration. In in vivo experiments, SA-2 microspheres implanted into rat cranial defects were able to promote neoangiogenesis, stimulate the proliferation of fibroblasts, and direct the migration of host cells to the defects and the interior of the microspheres, which ultimately promoted the formation of new bone [53]. In summary, β-TCP ceramics have many advantages in bone tissue engineering. By compounding with agarose hydrogel, it can also optimize mechanical properties, realize personalized customization, and better adapt to bone defect sites.

β-TCP/agarose scaffolds are highly promising for rapid osseointegration due to their accelerated degradation kinetics (8–12 weeks) and ability to release Ca^2+^ ions, which stimulate osteoblast activity and angiogenesis. Compared to HA/agarose composites, β-TCP exhibits faster resorption, making it ideal for temporary support in non-load-bearing defects. However, its low mechanical strength (<50 MPa) and rapid degradation risk premature structural collapse, limiting applications in load-bearing scenarios. Future research should focus on optimizing HA/β-TCP ratios to synchronize degradation with bone regeneration rates.

#### 2.1.3. Calcium Carbonate/Agarose Gels in Early-Stage Osteogenesis

Calcium carbonate (CaCO_3_) is one of the most abundant biomineralizers in nature and is considered a potential inorganic precursor for inducing the formation of bone minerals such as hydroxyapatite. Calcium carbonate does not cause significant immune responses in the body. Degradation products of calcium carbonate (e.g., calcium and carbonate ions) can participate in the mineralization process of bone tissue and promote the formation of new bone [54]. In 2010, Suzawa et al. investigated the regenerative behavior of biomineralized/agarose composite gels, including HA/agarose and CaCO_3_/agarose composite gels, prepared through an alternating immersion process, in cranial bone defects in rats. The HA and CaCO_3_/agarose composite gels were expected to accelerate the rate of new bone production associated with osteogenesis. It has an important role in biocompatible and biodegradable bone graft filler material as an alternative to autologous bone [55]. In 2015, Suzawa et al. formed HA or CaCO_3_ on agarose gels by an alternating immersion technique. The HA- and CaCO_3_-forming agarose gels significantly enhanced the osteogenic capacity of MSCs in the early stages and continued to support this capacity in the later stages. Specifically, bone-specific osteocalcin content was detected in CaCO_3_-forming agarose gels at day 14 and gradually increased over time. By contrast, HA-formed agarose gels detected similar osteocalcin content on day 21 and increased on day 28. By contrast, only a small amount of osteocalcin was detected in the untreated agarose gel. These results indicated that the agarose gels formed by HA and CaCO_3_ had a significant effect in supporting the osteogenic capacity of MSCs, and the agarose gels formed by CaCO_3_ showed a higher promotion in the early stage [56]. Under certain conditions, the bone formation ability of CaCO_3_ is even better than that of HA. If calcium carbonate, hydroxyapatite, and agarose can be reasonably combined, it is expected to achieve more excellent repair effect in the field of bone tissue engineering and provide more effective solutions for the treatment of bone defects.

#### 2.1.4. Chitosan/Agarose/HA Nanocomposites for Load-Bearing Applications

Additionally, the combination of chitosan and HA/agarose gel also shows great potential in bone tissue engineering. In 2019, Kazimierczak et al. prepared a novel chitosan/agarose/nano hydroxyapatite (chitosan/agarose/nanoHA) bone scaffold by a novel method combining freeze-drying and foaming agents. The scaffolds showed osteoinductive properties and were able to induce osteogenic differentiation (Runx2 synthesis) of undifferentiated mesenchymal stem cells (MSCs), and the surface was extremely hydrophilic and readily adsorbed proteins, especially with the highest adsorptive affinity for fibronectin, which contributes to osteoclast adhesion, spreading, and proliferation. Porous nanocomposite biomaterials can be a good solution to the limited cell migration and distribution and insufficient nutrient and oxygen transport [57]. Subsequently, Kazimierczak et al. investigated the effects of this highly macroporous scaffold on macrophage polarization and osteogenic differentiation. This biomaterial induces macrophage polarization to the M2 type, releasing high levels of anti-inflammatory cytokines (e.g., IL-4, IL-10, IL-13, and TGF-β1) that are promotive of osteogenic differentiation. The M2 type of macrophage has an immunomodulatory effect on the process of osteogenic differentiation through the secretion of anti-inflammatory cytokines, which results in the chit/agarose/HA scaffolds having a low risk of inflammation and being able to promote bone tissue repair and remodeling [58]. They also doped magnesium (Mg^2+^) and zinc (Zn^2+^) into nanohydroxyapatite of chitosan/agarose/hydroxyapatite and investigated the biological response of the porous bone scaffold. It was found that Mg^2+^-doped chit/aga/HA scaffolds significantly promoted the spreading and proliferation of osteoblasts and enhanced osteocalcin production by mesenchymal stem cells (MSCs). However, scaffolds made of pure HA performed better than Zn^2+^-containing materials. The experimental results clearly show that chit/aga/HA scaffolds modified with Zn^2+^ have no positive effect on cellular behavior, whereas doping with Mg^2+^ off ions may significantly improve the biocompatibility of the resulting materials and increase their potential for biomedical applications [59]. The effect of metal ions on the process of bone tissue regeneration suggests that the addition of these ions to existing bone substitution materials may modify inflammatory and foreign body responses or accelerate the onset of bone regeneration, as well as the durability of the materials [60]. In 2022, Chunling Yang et al. developed a novel agarose/gadolinium-doped hydroxyapatite (agarose/Gd-HA) composite bone filler material with a three-dimensional porous structure. It was found that agarose provided the three-dimensional skeleton and conferred porosity, processability, and high specific surface area; hydroxyapatite (HA) conferred biocompatibility; and the rare earth element gadolinium (Gd) acted as an antimicrobial agent. In this material, Gd was successfully doped into the HA lattice to form a Gd-HA interstitial solid solution, and there were chemical interactions between agarose and Gd-HA, and the Gd-doped HA modulated the physical structure of agarose. Significant bioactivity and osteogenic properties of the composite filler material were revealed using mouse osteoblast precursors (MC3T3-E1), and cell proliferation and growth rates increased with increasing Gd content in the composite. Antimicrobial tests using both *Staphylococcus aureus* (*S. aureus*) and *Escherichia coli* (*E. coli*) showed antimicrobial functionality, and the antimicrobial properties of the composites were enhanced with increasing Gd content [61].

Agarose–ceramic composites exhibit higher porosity than pure agarose hydrogels, resulting in better cell adhesion, showing great potential in addressing the mechanical and osteogenic needs of bone regeneration. HA/agarose excels in load-bearing applications due to its high compressive strength but suffers from slow degradation, potentially impeding remodeling in dynamic environments. In contrast, β-TCP/agarose scaffolds degrade rapidly and promote early osseointegration, yet their low mechanical resilience limits utility to non-load-bearing defects. CaCO_3_/agarose hybrids further enhance early-stage mineralization but lack structural stability. While chitosan/agarose/HA nanocomposites offer antimicrobial and osteoinductive synergy, their high porosity compromises mechanical durability under physiological stresses. A critical limitation across these systems is their static nature; they lack dynamic responsiveness to microenvironmental cues, a gap that necessitates the incorporation of bioactive molecular cues—such as growth factors, fibronectin, or magnetic nanoparticles—to achieve spatiotemporal control over tissue regeneration. This transition to dynamic, cell-instructive systems will be elaborated in Section 2.2, where agarose-based bioactive composites are explored for their ability to bridge the divide between structural support and biological responsiveness.

### 2.2. Agarose-Bioactive Molecular Composites

Agarose hydrogel is a weak hydrogel with the same spectral properties as unmodified agarose. Although AG exhibits many advantages in terms of biocompatibility and tissue response, there are potential drawbacks and limitations, such as variability in the thickness of the fibrous capsule and the density of neutrophils, differences in the neutrophil response, formation and degradation of the fibrous capsule, the mechanical properties of the material, and the rate of degradation of the material [62]. The use of bioactive composites can go a long way in compensating for the shortcomings of pure agarose hydrogels. These materials usually have good biocompatibility and low immunogenicity, which can provide a suitable growth environment for cells, reduce immune rejection, and promote cell attachment, proliferation, and differentiation. At the same time, they have controlled degradability and bioresorbability, which can match the growth rate of new bone tissue, gradually degrade, and be completely absorbed, avoiding the long-term problems that may be associated with permanent implants. In addition, bioactive composites are capable of releasing bioactive factors that induce the differentiation of stem cells toward osteoblasts, promote the proliferation and activity of osteoblasts, and accelerate the regeneration of bone tissue [63].

#### 2.2.1. Growth Factor-Loaded Agarose Hydrogels in Osteochondral Repair

Achieving the long-term goals of promoting fracture healing, facilitating bone–implant bonding, and increasing bone formation in patients with osteoporosis is dependent on the critical role of peptide growth factors. These growth factors are produced by osteoblasts and other bone cells in vitro and in vivo and are relatively abundant in the bone matrix. They stimulate the proliferation and differentiation of osteoblasts in vitro and promote bone formation when administered in vivo [64]. These properties give growth factors great potential for application in promoting fracture healing, enhancing bone–implant bonding, and increasing bone formation in patients with osteoporosis [65]. In addition, growth factors can stimulate the expression of cell surface adhesion molecules so as to improve the cell adhesion of the agarose complex. Specifically, platelet-derived growth factor (PDGF) activity is regulated by interactions with other growth factors (e.g., transforming growth factor beta, TGFβ) and pro-inflammatory cytokines. Insulin-like growth factors (IGFs) play an important role in the general growth and maintenance of the body’s skeleton; systemic application of IGFs increases bone formation and bone repair in animals but may also be accompanied by increased bone resorption and insulin-like systemic effects. Transforming growth factor β (TGFβ) acts as a potent growth inhibitor for many cell types, primarily as a morphogen, inducing matrix synthesis and angiogenesis in mesenchymal cells [66]. The combined effects of these growth factors provide a powerful tool for orthopedic medicine with the promise of improved fracture treatment and implant integration.

Agarose hydrogels can support the survival and chondrogenic differentiation of human adipose-derived adult stem (hADAS) cells and promote the synthesis and accumulation of cartilage matrix macromolecules [67]. However, its cell biosynthesis rate is low, and its mechanical properties perform poorly compared to other hydrogels [68,69]. To ameliorate this problem, in 2004, Awad et al. investigated the chondrogenic differentiation of hADAS in agarose, alginate, and gelatin scaffolds. Chondrogenic medium-containing transforming growth factor β1 significantly increased the rate of protein and proteoglycan synthesis, as well as the DNA, chondroitin sulfate, and hydroxyproline content of the engineered constructs. The increase in shear modulus was significantly correlated with the increase in chondroitin sulfate content (R^2^ = 0.36; *p* < 0.05) and the interaction between chondroitin sulfate and hydroxyproline (R^2^ = 0.34; *p* < 0.05) [25]. Huang et al. used agarose gel as a support material for chondrogenesis of human bone-marrow-derived mesenchymal stem cells hBM-MSCs. Chondrogenesis was induced in hBM-MSCs by culturing the cell–agarose constructs and pellet in medium containing transforming growth factor β3 (TGF-β3) for 21 days, and chondrogenesis was successfully expressed with the chondrogenic markers type II collagen and aggregated proteoglycans. tGF-β3 induced chondrogenic differentiation of hBM-MSCs, promoted the expression of cartilage-specific markers, influenced extracellular matrix synthesis and deposition, facilitated the formation of cellular aggregates, and increased the efficiency of chondrogenesis. In addition, chondrogenesis in agarose culture was directly correlated with the initial cell inoculum density, and cell-agarose constructs with high initial cell inoculum densities exhibited more chondrogenic-specific gene expression [70]. In 2006, Mauck et al. investigated the chondrogenic differentiation and functional maturation of bovine bone marrow-derived mesenchymal stem cells (MSCs) in long-term agarose cultures. MSCs were able to undergo chondrogenesis in agarose, but the amount of matrix formed and the mechanical properties were lower than those produced by chondrocytes under the same conditions. It is worth noting that some important properties in the MSC constructs, such as glycosaminoglycan content and equilibrium modulus, leveled off over time, suggesting that the decrease in their ability was not simply the result of delayed differentiation [71]. Although MSCs are capable of generating constructs with significant cartilage properties, further optimization is needed to achieve a level similar to that of chondrocytes.

In order to focus on the effect of mechanical stimulation on cell behavior and to avoid the possible adverse effects (e.g., fibrosis and heterotopic ossification) that may be caused by high concentrations or long-term use of growth factors such as TGF-B, the researchers conducted experiments in which, by dynamically loading by simulating the mechanical loads in the physiological environment, the method was able to significantly promote matrix synthesis and tissue development of chondrocytes and improve the mechanical properties of the engineered cartilage making it closer to the properties of natural cartilage [72]. Finger et al. showed that circulating hydrostatic pressure (CHP), in the absence of transforming growth factor beta (TGF-b), induced chondrogenic differentiation of bone-marrow-derived hMSCs in agarose constructs. Either progressively increasing or stabilizing CHP initiated this process, and higher initial pressures may trigger chondrogenesis more rapidly [73]. In 2013, a study by Puetzer et al. further found that CHP was able to upregulate the mRNA expression of Sox9, aggrecan, and COMP in hASCs, suggesting that CHP can initiate chondrogenic differentiation in hASCs, similar to hMSCs. However, by day 14, loaded hASC constructs showed lower mRNA expression of chondrogenic markers than unloaded controls, and by day 21, the samples showed almost undetectable mRNA expression and significantly reduced cell activity. These results suggest that 2% pure agarose hydrogel may not be able to support the activity of hASC or hMSC for a long time in a complete growth medium without chondrogenic inducers but by CHP induction can repair bone defects well [74]. These studies confirm the ability of mechanical stimulation alone to induce hASC chondrogenic differentiation in the absence of chondrogenic factors, emphasizing the importance of considering mechanical stimulation and appropriate three-dimensional culture when using hASCs for cartilage tissue engineering.

Growth factor-loaded agarose hydrogels (e.g., TGF-β, IGF, PDGF) exhibit remarkable potential in osteochondral repair by enhancing stem cell differentiation and extracellular matrix synthesis, while mechanical stimulation—such as CHP—synergistically promotes matrix deposition and improves mechanical properties. However, pure agarose hydrogels fail to sustain mesenchymal stem cell (MSC) activity over extended periods in the absence of growth factors, underscoring the necessity of hybrid strategies. Future advancements could combine growth factor delivery with mechanical stimulation protocols to reduce reliance on supraphysiological factor concentrations, thereby minimizing risks of fibrosis or heterotopic ossification associated with overdosing.

#### 2.2.2. Fibronectin–Agarose Hydrogels for Cartilage Regeneration

In addition to growth factors, the binding of fibronectin to agarose has been shown to have potential applications in tissue engineering. Fibrous proteins, especially collagen, play a crucial role in bone tissue engineering [75]. Collagen, as a major structural component of tissues, not only provides significant tensile strength and cell attachment sites to promote cell attachment but also supports neovascularization and nerve regeneration, thus enhancing tissue integration and functional recovery [76]. In 2020, Cambria et al. developed and evaluated a novel agarose–collagen composite hydrogel. This composite hydrogel was prepared by physical mixing, forming a homogeneous polymer network, and the mechanical properties of the hydrogel were similar to those of pure agarose at a collagen concentration of 2 mg/mL. The composite hydrogel showed significant advantages in promoting cell adhesion, enhancing extracellular matrix (ECM) gene expression, and increasing glycosaminoglycan (GAG) production compared to collagen-free agarose hydrogel. In addition, composite hydrogels showed higher pyrophosphorylated tyrosine 397 expression of adhesion patch kinase (pFAK) under dynamic compression, contributing to a deeper understanding of cell–matrix interactions and mechanotransduction mechanisms [77].

Laura García-Martínez et al. investigated the successful encapsulation and packaging of human elastic cartilage-derived chondrocytes (HECDC) in nanostructured fibronectin-agarose hydrogels (NFAH) and the ability to proliferate and form clusters of cells expressing S-100 and waviness proteins during in vitro culture, synthesizing different extracellular matrix (ECM) molecules, including type I and type II collagen, elastic fibers, and proteoglycans. HECDC in NFAH showed higher ECM synthesis capacity compared to alginate oxide hydrogel matrix alone. NFAH can be used to generate biodegradable and bioactive constructs suitable for cartilage tissue engineering applications [34]. In 2018, Fernando Campos et al. utilized curcumin (genipin)-crosslinked fibrin–agarose hydrogel and nanostructured fibrin–agarose hydrogel (FAH and NFAH) to organize similar models for tissue engineering applications. Curcumin cross-linking significantly improved the structural and biomechanical properties of FAH and NFAH. Curcumin cross-linking increased the stiffness (G) and elasticity (G′) of the hydrogels, especially at a curcumin concentration of 0.75%; the G value of FAH increased from 31.5 Pa to 470 Pa, and that of NFAH increased from 500 Pa to 7000 Pa. Curcumin cross-linking decreased the porosity of the hydrogels from 65.27% to 41.63% for FAH. The porosity of NFAH was also reduced. However, higher concentrations of curcumin (0.75%) began to negatively affect cell function and activity. Natural and biocompatible FAH and NFAH with improved structural and biomechanical properties can be generated using 0.1% to 0.5% curcumin, but further in vivo studies are needed to validate the biocompatibility, biodegradability, and regenerative capacity of these cross-linked scaffolds [78]. In conclusion, these investigations have demonstrated that the crosslinking technique of nanostructured fibronectin–agarose hydrogel may be optimized to greatly increase its application potential in cartilage tissue creation. Additional in vivo research will help to confirm its true effect.

#### 2.2.3. Magnetic Nanoparticle-Embedded Agarose for Guided Tissue Assembly

Magnetic nanoparticles (MNPs) are capable of precisely guiding cellular localization and assembly through external magnetic fields to form complex three-dimensional tissue structures such as blood vessels, nerves and muscles. It enables better control of cell guidance, release of bioactive factors, and tissue maturation [79]. 2020, Ana Belén Bonhome-Espinosa et al. investigated a novel magnetic fibrin-agarose hydrogel (3D-MCFAC) for cartilage tissue engineering. In the study, magnetic nanoparticles and human hyaline chondrocytes were encapsulated in a fibronectin–agarose hydrogel to generate articular cartilage-like tissue. Rheological measurements revealed that the incorporation of magnetic nanoparticles significantly increased the energy storage modulus and loss modulus of the hydrogels and exhibited higher values at different time points of cell culture. Hydrogels encapsulating human hyaline chondrocytes were able to control their swelling capacity compared to cell-free magnetic and non-magnetic hydrogels. Interestingly, in vitro cell activity and proliferation results showed that the incorporation of magnetic nanoparticles did not affect the cytocompatibility of the biomaterials. Also, magnetic nanoparticles did not negatively affect type II collagen expression in human hyaline chondrocytes. Further studies are needed to elucidate whether human hyaline chondrocytes in magnetic fibronectin–agarose hydrogels are able to express more features typical of the cartilage extracellular matrix [80].

In 2023, Mokhtarzade et al. developed a gradient four-layer gelatin methacrylate (GelMA)/agarose structure as an injectable scaffold for mimicking osteochondral (OC) tissue. By adjusting the ratio of GelMA and agarose, an injectable four-layer scaffold was designed in which the lower and upper layers simulated the extracellular matrix (ECM) of bone and cartilage, respectively. The porosity of each layer ranged from 76% to 96%, with an average pore size of approximately 115 µm. During degradation, the weight loss process was uniform for all layers, and the average residual weight of each layer after four weeks ranged from 48% to 58%. The compressive modulus of the layers ranged from 12 to 76 kPa, and the layers retained their structure at about 40% and 70% strain at 10% *w*/*v* and 15% *w*/*v* compositions, respectively. In addition, the layers exhibited appropriate injectability with viscosity values ranging from 13.05 to 68.19 Pa.s and a torque of 50%. The cellular activity was higher than 91% and 86% at 1 and 7 days after injection, respectively. The results suggest the potential of this scaffold as an injectable, gradient structure for future OC tissue engineering applications [81].

By introducing bioactive molecules and composite materials, the performance of agarose hydrogels has been significantly improved, opening up new ideas and strategies for cartilage and bone tissue engineering. However, traditional methods have limitations in accurately manufacturing complex three-dimensional structures, especially in simulating the porosity and microscopic characteristics of natural tissues. The next chapter will introduce an emerging injectable agarose processing strategy to solve this problem.

## 3. Additive Manufacturing and Bio-3D Printing

Three-dimensional printing technology, also known as additive manufacturing, involves the use of computer-aided design software to create a digital model and 3D printers to slice this model into thin layers to form a physical item by stacking materials layer by layer [82]. Three-dimensional bioprinting is the combination of 3D printing and biomaterials, whereby biomaterials are deposited layer by layer through precise computer control of bio-inks to form biocompatible bionic structures, which have high prospects for application in tissue engineering, drug delivery, and other areas [83]. Based on the working principle, 3D bioprinting is mainly divided into four different types: extrusion, inkjet, laser-assisted, and stereolithography [84] (Figure 3).

The key to bio-3D printing lies in the selection of bio-inks. The ideal bio-ink not only requires meeting the requirement of maintaining stable mechanical properties during the printing process but should also have good biocompatibility and cytocompatibility when implanted into the human body or loaded into cells [63]. Different printing methods have different requirements for bioinks; Table 1 summarizes the common bio-3D printing methods, which include advantages, disadvantages, and required bioink properties.

While injectable agarose hydrogels are ideally suited to be used as a platform for cell inoculation, traditional injection methods have limitations in achieving complex geometries and fine features [96]. By contrast, 3D printing technology overcomes this challenge by enabling rapid fabrication of hydrogels with complex structures.

### 3.1. Injectable Agarose Bioinks: Printability

Printability refers to a bioink’s capacity for precise deposition and its ability to produce printed constructs with the intended shape and structure when subjected to specific printing conditions. It involves a number of key factors including viscosity, shear thinning, yield stress, elastic recovery, cell loading, and cross-linking properties [97]. Bioinks with lower viscosity extrude more smoothly but cause the 3D structure to become softer, making it difficult to maintain the printed shape, and the cells are susceptible to shear forces during the printing process, leading to cell damage or death [98]. Shear-thinning materials have a viscosity that decreases with increasing shear rate when subjected to shear. As the bioink passes through the print nozzle, the shear-thinning material is able to be extruded at a lower pressure, reducing the mechanical damage to the cells. Once the shear force disappears, the viscosity of the material quickly recovers, maintaining the shape and stability of the printed structure [99]. Injectable agarose bioink is one of the commonly used materials for bio-3D printing due to its low temperature, fast gelation, and shear-thinning properties, and its ability to carry live cells [100]. Although agarose hydrogels have excellent gelation properties, further optimization and tuning of agarose hydrogels is needed to better suit the needs of bio-3D printing.

Since the viscosity of agarose bioinks depends on the agarose concentration, printability can be improved by controlling the agarose concentration. In 2022, wenger et al. found that agarose concentration was positively correlated with viscosity. The viscosities of 3%, 4.5%, and 6% agarose bioinks were approximately 100 mPa-s, 200 mPa-s, and 300 mPa-s, respectively, at 70 °C. Upon lowering the temperature to the nozzle temperature, the 3% and 4.5% agarose bioinks exhibited shear thinning behavior [101]. However, as the agarose concentration increases, the porosity of the hydrogel decreases, affecting metabolic waste drainage and nutrient uptake by the cells, thereby adversely affecting cell survival [102]. In 2017, Forget et al. developed a thermosensitive bioink based on carboxylated agarose (CA). By regulating the degree of carboxylation of CA, precise modulation of the elastic modulus of the printing gel in the range of 5–230 Pascal was achieved, while the shear viscosity of the bioink aqueous solution was barely altered (10–17 mPa). CA causes less cellular damage during cell proliferation and better protects the cells from damage caused by shear forces and high agarose concentrations. This unique property enables the fabrication of 3D structures composed of different mechanical domains with the same printing parameters and low nozzle shear stress [103]. The viscosity of bioinks is critical to their printing performance; however, cell viability is also a key consideration when selecting bioinks.

In bio-3D printing, bioinks should not only be able to maintain high cell viability during and after printing but also promote cell proliferation or differentiation [104]. This requires bioinks to have excellent cell-loading capabilities. Functionalizing agarose hydrogels is an effective way to enhance their cell adhesion. In 2019, Arya et al. developed an extrudable carrier based on RGDSP-functionalized carboxylated agarose for chondrocyte delivery. It was found that the stiffer hydrogel (5.8 kPa) was more effective in promoting chondrogenesis than the softer hydrogel (0.6 kPa). In the GGGGRGDSP-modified hydrogel, the synergistic effect of hardness and RGD signaling significantly enhanced the expression of genes associated with chondrogenesis, including aggrecan, collagen type II, and sox9. This functionalized carboxylated agarose hydrogel demonstrated enhanced bioactivity in in vitro experiments and was able to promote chondroitin sulfate (s-GAG) secretion. In in vivo experiments, the hydrogel was able to support the chondrogenic phenotype of human articular chondrocytes (HACs), which in turn promoted the formation of new cartilage [105].

Bone engineering, as an important branch of regenerative medicine, places more stringent demands on the performance of biomaterials. Next, we will explore in detail the 3D printing technology of agarose hydrogels for bone engineering and its potential and challenges in constructing biologically active and mechanically supported bone tissue substitutes.

### 3.2. Three-Dimensional Printing of Agarose Hydrogels for Bone Engineering

When the size of a bone defect exceeds a critical threshold, the natural healing ability of the bone is drastically reduced, making the healing process extremely difficult or even impossible to repair on its own [106]. At this point, scaffolds fabricated using 3D printing technology can effectively aid in bone tissue repair. Currently, a variety of biomaterials, including agarose hydrogel, have been widely used in the field of bone tissue engineering [8,84,107,108]. Adequate blood vessel supply plays a key role in the process of bone tissue regeneration. It has been found that cartilage growth is often inhibited by the lack of sufficient oxygen and nutrients to the cells in a three-dimensional culture system. The oxygen consumption rate of agarose was twice that of collagen, suggesting that agarose hydrogels have a stimulating effect on cell metabolism. In addition, the rate of degradation of the agarose hydrogel scaffold should be controlled to ensure that it matches the rate of bone tissue regeneration. The scaffold should gradually degrade during the gradual formation of bone tissue and eventually be completely replaced by newborn bone tissue [109]. Table 2 summarizes several representative agarose bio-ink formulations and summarizes the relevant printing methods and research parameters used to evaluate printability. The crosslinking methods of these bio-inks and their application functions in bone repair are also introduced.

Fedorovich et al. improved the biocompatibility and cell survival of the scaffolds by printing bone marrow mesenchymal stem cells (BMSCs) mixed with different concentrations of hydrogel via 3D fiber deposition system. The cells were able to survive the extrusion process, and their subsequent survival was not significantly different from that of unprinted cells, and the BMSCs were able to differentiate along the osteoblast lineage after extrusion [116]. This suggests that cell-carrying scaffolds printed using this technique have potential applications in bone tissue engineering. Duarte Campos et al. found that osteogenic differentiation was preferentially achieved in anisotropic soft collagen-rich matrices, whereas adipogenic differentiation occurred predominantly in anisotropic hard agarose-rich matrices. Using bioactive type I collagen–agarose bioinks, hydrogels with different structural and mechanical properties were created by adjusting the ratio of agarose to type I collagen, resulting in improved scaffold printing performance and cell differentiation potential [114]. Arya et al. on the other hand tuned the internal cross-linking and mechanical properties of the scaffolds by the carboxylation of agarose, thereby improving the printability and structural retention of the scaffolds. In scaffolds containing the integrin-binding peptide sequence GGGRGDSP, the synergistic effect between scaffold stiffness and RGD signaling resulted in enhanced expression of chondrogenesis-related genes (e.g., aggrecan, collagen type II, and sox9). The addition of carboxylation increased cell adhesion and proliferation relative to unmodified agarose hydrogels. However, the addition of carboxylation also produced a denser scaffold network and faster swelling and degradation in water over a longer period of time, which may reduce the space for cell growth and proliferation [105]. Therefore, when utilizing agarose-based hydrogels as bioinks, it is important to consider their mechanical properties during printing and the associated performance after printing, as well as to ensure that they are effective in promoting cell expression after implantation into the human body.

When bioink passes through a nozzle, the phenomenon of orientation and storage of elastic energy occurs at the outlet of the pipe or nozzle due to the viscous and elastic properties of the fluid. This phenomenon can cause the shape and size of the fluid at the outlet to deviate from the expected design, thus affecting the shape fidelity in precision manufacturing processes such as 3D printing [117]. Wang et al. used a thermally assisted 3D printing method combining the use of agarose and calcium alginate (CA/Ag) to minimize the Barus effect and to further improve the print resolution. The introduction of agarose changed the rheological properties of the ink, resulting in 3D-printed structures with higher accuracy. In addition, the introduction of a soft polyacrylamide (PAAm) network into the 3D-printed CA/Ag hydrogel, combining a rigid calcium alginate network with a soft PAAm network, enhanced the interfacial surfaces between neighboring stripes, thus improving the mechanical properties of the scaffolds [118]. In summary, agarose-based hydrogels have a broad application prospect as bioinks in 3D printing, but their application process requires comprehensive consideration of multiple factors. It is necessary to pay attention to the mechanical properties of bioink during the printing process, such as the effect of fluid viscosity and elasticity on the fidelity of printed shapes, as well as the relevant performance after printing, such as cell adhesion, proliferation, and the expression of chondrogenesis-related genes. It is also necessary to consider whether cell expression can be effectively promoted after implantation into the human body, and how to optimize the performance of the scaffolds by adjusting the ratio of agarose to other components or modifying them.

## 4. Summary and Outlook

Agarose-based hydrogels have significant applications in the field of bone defect repair due to their unique thermoreversible gel properties, excellent mechanical properties, good biocompatibility, and economical cost [20]. By compounding with ceramic materials, bioactive molecules, etc., the properties of agarose hydrogels have been further enhanced to better meet the needs of bone tissue engineering. For example, compounding with hydroxyapatite not only enhanced the cross-linking degree and mechanical properties of the gel but also significantly improved its osteogenic capacity [42,51,52]. Furthermore, the application of agarose hydrogels as bioinks in bioprinting technology offers the possibility of constructing complex three-dimensional bone tissue structures. However, researchers have used a variety of strategies, such as chemical modification, mixing with other materials, and adding growth factors to solve the problems of insufficient cell adhesion, high rigidity, and low surface roughness of agarose hydrogels, which limit their effectiveness in some applications [119]. Future research can further explore the application of nanotechnology, such as the uniform dispersion of nano-ceramic particles and the addition of nanofibers, in order to further improve the mechanical properties and biological functions of agarose hydrogels. At the same time, gene editing technology can be combined to accurately regulate the behavior and fate of cells in agarose hydrogels, so as to achieve more accurate bone tissue engineering. On the other hand, the development of bioprinting technology will bring new opportunities for the application of agarose hydrogels [18]. The optimization of printing parameters and the development of novel bioinks are expected to enable the construction of more complex and precise bone tissue structures that better mimic the structure and function of natural bone tissue. The clinical transformation of additive manufacturing and biological 3D printing technology needs to take multi-material gradient printing as the core breakthrough point and simulate the mechanical heterogeneity of natural bone by developing multi-nozzle system integration. On this basis, intelligent bioinks can be designed, such as photocurable agarose and temperature-sensitive GelMA composite systems, which can be triggered by ultraviolet light to achieve dynamic regulation of modulus, and the controlled release of growth factors through stimuli-responsive nanocarriers to adapt to the dynamic changes in the damage microenvironment. Combined with the thermal response characteristics of 4D printing [120], the precise bone regeneration of ‘printing as treatment’ in a minimally invasive environment can finally be realized, opening up a new dimension for personalized medical treatment.

## Figures and Tables

**Figure 1 gels-11-00255-f001:**
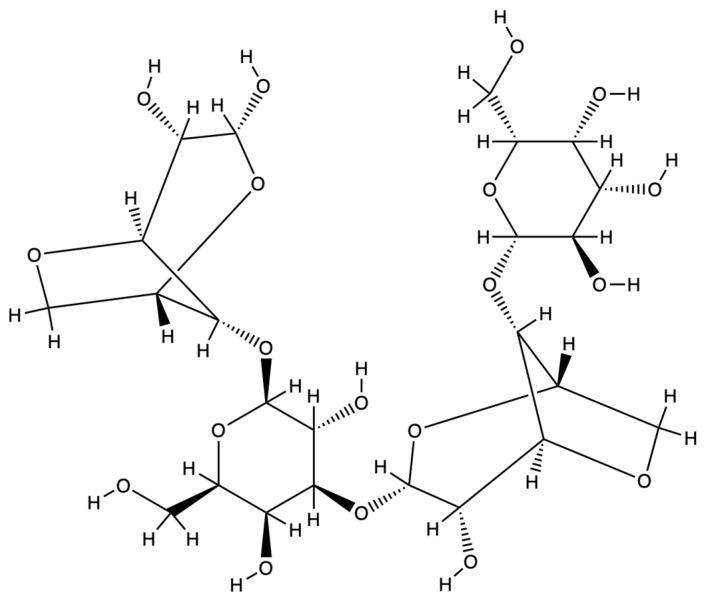
The molecular structure of agarose.

**Figure 2 gels-11-00255-f002:**
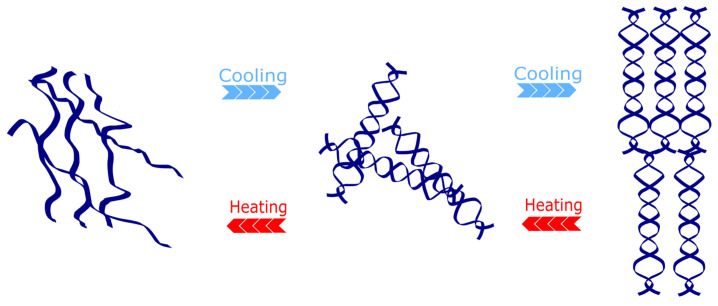
Structural changes in agarose hydrogels during heating and cooling.

**Figure 3 gels-11-00255-f003:**
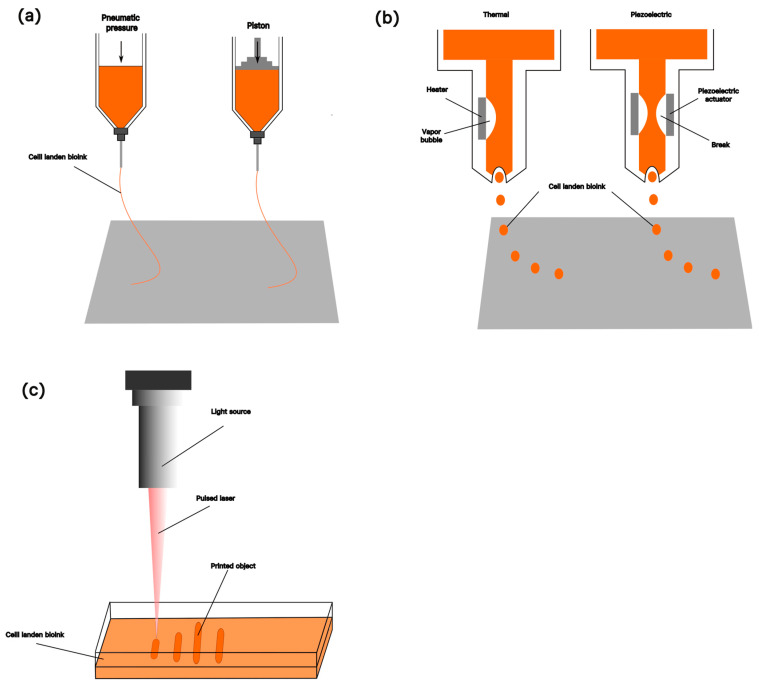
The technology used in 3D printing of cell biology at the present stage: (**a**) extrusion biological 3D printing driven by air pressure and driven by piston machinery; (**b**) ink-jet 3D bioprinting, which jets bio-ink by heating to generate steam bubbles and controls ink jets through the piezoelectric effect; (**c**) photocuring 3D bioprinting technology.

**Table 1 gels-11-00255-t001:** Advantages, disadvantages, and required bioink properties for hydrogel bio-3D printing.

Printing Method	Subcategories	Vantage	Drawbacks	Accurate	Cell Viability	Required Bio-Ink Properties	Bibliography
Extrusion printing	Pneumatic extrusion and mechanical extrusion	Simple and economical to operate; fast printing speeds; ability to print bio-inks with high cell densities	Cells may experience higher shear stresses leading to damage; specific requirements for rheological properties of bioinks; additional post-processing steps required; difficulty in achieving very high resolution.	30 µm–1 mm	40–80%	Viscosity needs to be high enough to maintain the shape of the printed structure, but not so high that it cannot be extruded smoothly. Viscosity in the range of 30 to 6 × 10^7^ mPa-s; moderate gel speed to quickly stabilize structures and protect cells	[18,85,86,87]
Inkjet printing	Continuous Inkjet; Drop-on-Demand	High resolution; non-contact printing; high cell viability; drop-on-demand feature reduces material waste	Uses small nozzle diameters that are prone to clogging and require regular maintenance and cleaning; printing equipment is relatively complex; and the viscosity of the bio-inks used must be low	10 µm–500 µm	70–90%	Lower viscosity for smooth spraying through the nozzle, viscosity range: 3–30 mPa-s; faster gelling speed, reducing droplet spreading on the print substrate; rheology	[88,89,90,91]
Light curing	Stereolithography (SLA); digital light processing (DLP)	High resolution; fast curing; to use a wide range of materials; microfluidic channels can be printed	Requires addition of photocrosslinking agents; expensive; ultraviolet (UV) light may be phototoxic to cells; requires removal of support structures	12 µm–150 µm	>80%	Rapid cure reduces distortion and collapse; viscosity range: 100–1000 mPa-s; shear thinning and thixotropy	[92,93,94,95]

**Table 2 gels-11-00255-t002:** Examples of the application of 3D bioprinting agarose-based hydrogels in tissues engineering.

Printing Method	Ink Composition	Crosslinking Method	Function	Bibliography
Fused deposition modeling	200 mg agarose, 200 mg sodium alginate, 89.7 mg Irgacure 2959, 2.84 g acrylamide (AAm), 1.85 mg N,N′-methylene bis(acrylamide)	Optical crosslinking; physical crosslinking	Extremely tough, self-recoverable, high shape fidelity	[110]
Extrusion-based 3D printing	4% agarose; type VII	Physical crosslinking	Support the phenotypic differentiation of MSCs into hyaline cartilage	[111]
Extrusion-based 3D printing	5% *w*/*v* agarose-alginate mixture with a 3:2 ratio of agarose to alginate.	Physical crosslinking	No need to add sacrificial materials	[112]
Extrusion-based 3D printing	5% *w*/*v* agarose-alginate mixture with a 3:2 ratio of agarose to alginate	Physical crosslinking	Accelerated cell proliferation; improve compression stiffness and tensile strength	[113]
Extrusion-based 3D printing	2% *w*/*v* RGDSP functionalized carboxylated agarose	Physical crosslinking	Supporting chondrogenic differentiation; adjustable mechanical properties	[105]
Inkjet bioprinting	Hydrogel mixture of 0.5 g/mL agarose and 0.21 g/mL	Physical crosslinking	Change the mechanical stiffness and print profile of the printed structure	[114]
Extrusion-based 3D printing	Agarose: 1.5 g in 100 mL of 1× PBS Graphene Oxide: 1.0% (*w*/*v*) solution Hydroxyapatite: 4.0% (*w*/*v*) solution	Physical crosslinking	Enhanced osteoinductive behavior, cellular differentiation	[115]
Fused Deposition Modeling	5% *w*/*v* agarose	Physical crosslinking	Good primary cell survival	[116]

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
