# Peer review of "Agarose Hydrogels for Bone Tissue Engineering, from Injectables to Bioprinting"

_gels, 2025, doi:10.3390/gels11040255_

Round 1

Reviewer 1 Report

Comments and Suggestions for Authors

This well written review touches the application of agarose as a biomaterial for bone restructuring and healing. It provides sound informatio, a comprhensive list of references and raises the interest of even not directly involved scientists.
This reviewer would however appreciate some additional restructuring of the manuscript as well as information.

1. Adding some subtitles which hint to the use of agarose-hydrogels in specific applcations would be helpful as a follow-up for the content.
2. In the introduction the authors refer to properties of agarose in the regulation of inflammatory reponses, etc. A short hint of underlying meachanisms would help for a better reasoning.
3. It is well known from biomaterial research, that polysaccharides (agarose is a PS)  activate the complement system due to their OH-groups. A short hint to this disadvantage is recommended.
4. Please correct beginning of sentence on page 15, line 534.
5. This reviewer finally congratulates the authors for a comprehensive review. 

Author Response

1.The reviewer’s comment:

Adding some subtitles which hint to the use of agarose-hydrogels in specific applcations would be helpful as a follow-up for the content...

The authors’ answer: 

To enhance the clarity and application-focused narrative, we have added specific subtitles throughout parts 2.1 and 2.2 to highlight the use of agarose hydrogels in targeted bone tissue engineering applications. Key modifications include:

2.1.1. Hydroxyapatite/Agarose Composites for Bone Defect Repair,

2.1.2. β-TCP/Agarose Scaffolds for Rapid Osseointegration,

2.1.3. Calcium Carbonate/Agarose Gels in Early-Stage Osteogenesis,

2.1.4. Chitosan/Agarose/HA Nanocomposites for Load-Bearing Applications,

2.2.1. Growth Factor-Loaded Agarose Hydrogels in Osteochondral Repair,

2.2.2. Fibronectin-Agarose Hydrogels for Cartilage Regeneration,

2.2.3. Magnetic Nanoparticle-Embedded Agarose for Guided Tissue Assembly.

2.The reviewer’s comment:

In the introduction the authors refer to properties of agarose in the regulation of inflammatory reponses, etc. A short hint of underlying meachanisms would help for a better reasoning.

The authors’ answer:

Based on your feedback, we have added a brief description of the role and mechanism of agarose hydrogel in reducing inflammatory responses in Chapter 2.

3.The reviewer’s comment:

It is well known from biomaterial research, that polysaccharides (agarose is a PS)  activate the complement system due to their OH-groups. A short hint to this disadvantage is recommended.

The authors’ answer:

We have added a discussion on the possible complement activation by agarose’s hydroxyl groups in Section 2 ("Agarose Hydrogel") .

4.The reviewer’s comment:

Please correct beginning of sentence on page 15, line 534.

The authors’ answer:

The sentance has been improved.&nbsp

5.The reviewer’s comment:

This reviewer finally congratulates the authors for a comprehensive review..

The authors’ answer:

We sincerely thank the reviewer for their kind acknowledgment and encouragement.

Reviewer 2 Report

Comments and Suggestions for Authors

The reviewed publication provides a comprehensive overview of agarose hydrogels in bone tissue engineering, covering both material properties and bioprinting technologies. However, several minor issues were identified. Although numerous studies are presented, the paper lacks a critical comparative analysis of different methods and materials, which would better highlight the advantages and disadvantages of each approach. While the authors mention issues related to cell adhesion and hydrogel stiffness, they do not detail the research efforts undertaken to address these challenges. Additionally, although the publication is a review, it lacks specific recommendations for the practical application of the discussed solutions. Despite these minor shortcomings, the publication offers valuable insights into the field.

Author Response

1.The reviewer’s comment:

The reviewed publication provides a comprehensive overview of agarose hydrogels in bone tissue engineering, covering both material properties and bioprinting technologies. However, several minor issues were identified. Although numerous studies are presented, the paper lacks a critical comparative analysis of different methods and materials, which would better highlight the advantages and disadvantages of each approach. While the authors mention issues related to cell adhesion and hydrogel stiffness, they do not detail the research efforts undertaken to address these challenges. Additionally, although the publication is a review, it lacks specific recommendations for the practical application of the discussed solutions. Despite these minor shortcomings, the publication offers valuable insights into the field.

The authors’ answer:

Heartfelt thanks to the reviewers for their careful review and constructive comments and the affirmation of the review. We have made the following changes to reviewer’s proposal:

1、The paper lacks a critical comparative analysis of different methods and materials

To enhance the clarity and critical analysis of the manuscript, in the 2.1 and 2.2 chapters, we added subheadings and added the advantages and limitations of agarose hydrogel composite with different materials at the end of the paragraph, so that readers can better understand its characteristics.

2、The research work on the cell adhesion and hydrogel stiffness was not described in detail.

In chapters 2.1 and 2.2, it is more clearly explained how ceramic / agarose hydrogel and growth factor / agarose hydrogel affect cell adhesion and hydrogel stiffness.

3、It lacks specific recommendations for the practical application of the discussed solutions.

At the end of the review, the future development and application of agarose hydrogel were discussed, and the discussion of its complex was also added in many parts of the review.